# Susceptibility Tests and Predictions of Transporter Profile in *Serratia* Species

**DOI:** 10.3390/microorganisms10112257

**Published:** 2022-11-14

**Authors:** Gunther J. Staats, Samantha J. Mc Carlie, Charlotte E. Boucher-van Jaarsveld, Robert R. Bragg

**Affiliations:** Department of Microbiology and Biochemistry, University of the Free State, 205 Nelson Mandela Drive, Park West, Bloemfontein 9301, South Africa

**Keywords:** disinfectant resistance, efflux pump, reserpine, carbonyl cyanide 3-chlorophenylhydrazone, *Serratia marcescens*, multidrug-resistance

## Abstract

Disinfectants and biosecurity are critically important to control microbial diseases. Resistance to disinfectants compromises sectors such as agriculture and healthcare systems. Currently, efflux pumps are the most common mechanism of antimicrobial resistance. This study aimed to identify the efflux transporters responsible for disinfectant resistance in a multidrug-resistant isolate *Serratia* sp. HRI compared to a susceptible *Serratia* sp. type strain. An efflux system profile was generated using the Transporter Automatic Annotation Pipeline (TransAAP) for both isolates. Thereafter, the efflux pump inhibitors, reserpine (RSP) and carbonyl cyanide 3-chlorophenylhydrazone (CCCP) were used to reveal the role of efflux pumps in susceptibility to three disinfectants (Didecyldimethylammonium chloride, HyperCide^®^, and benzalkonium chloride). Interestingly, the resistant isolate had fewer efflux systems in total compared to the type strain and fewer efflux systems classified as resistance efflux pumps. After the addition of RSP, a significant reduction in resistance capabilities against all three antimicrobials was observed for both isolates. However, CCCP supplementation produced mixed results with some outcomes suggesting the involvement of the Eagle effect. This study provides evidence that efflux pumps are responsible for the disinfectant resistance phenotype of the *Serratia* species due to the increased susceptibility when efflux pump inhibitors are added.

## 1. Introduction

The *Serratia* genus is categorized as part of the *Enterobacteriaceae* family, and members are commonly found in soil and water associated with plants, insects, animals, and humans [1]. *Serratia marcescens* is most frequently associated with human infections [2]. Clinical isolates harbor intrinsic chromosomal and acquired plasmid-encoded elements conferring resistance toward many antibiotics and disinfectants [1].

Incorporating disinfectants as a preventative measure, in combination with antibiotic administration, has become an integral part of infection control. However, the rising threat of antibiotic and multidrug resistance accompanied by opportunistic pathogens is a worldwide health concern [3]. This rise of untreatable/resilient bacterial populations increases the need for effective sterilization/disinfection methods that prevent the proliferation and spread of resistance [4]. Resistance against quaternary ammonium compounds (QACs) has been reported in clinical, industrial, and veterinary environments, and resistance genes specific to QACs have been identified [5]. One of the best-studied modes of disinfectant resistance is efflux systems in the small multidrug resistance (SMR) subfamily and major facilitator superfamily (MFS) [6,7]. These systems consist of cytoplasmic membrane-embedded multidrug transporters that recognize and expel toxic compounds inside the cell to the external environment [8].

In this study, the genomes of *S. marcescens* subspecies *marcescens* strain ATCC 13880 and multidrug-resistant isolate *Serratia* species Highly Resistant Isolate (HRI) [9] were submitted to a prediction pipeline tool to compare efflux system profiles and identify any resistance efflux systems [10]. Susceptibility testing was conducted using three test disinfectants against each *Serratia* isolate and two efflux pump inhibitors (EPIs) were added to determine efflux pump activity. Two standard EPIs were chosen, reserpine (RSP) and carbonyl cyanide 3-chlorophenylhydrazone (CCCP). RSP inhibits efflux by binding specific amino acid residues on the efflux pump, causing inhibition in certain efflux families, including the MFS, resistance-nodulation-division (RND) superfamily, and ATP-binding cassette (ABC) superfamily [11,12]. CCCP is classified as a protonophore as it disrupts the proton motive force (PMF) requirements of efflux pumps of the secondary transporter classification [13,14]. This inhibitor affects efflux systems belonging to the MFS, RND, drug/metabolite transporter (DMT), and multidrug/oligosaccharidyl-lipid/polysaccharide flippase (MOP) superfamilies [13,14].

This study aimed to first generate a list of predicted transporters in each *Serratia* isolate. Secondly, from the predicted transporters, individual transporters indicating resistance function were grouped, specifically grouping functionality/substrate specificity for QAC and multidrug resistance. Thirdly, the susceptibility/resistance profiles of the *Serratia* isolates were determined using test disinfectants. Finally, this work revealed a possible link between the resistance expressed and transporters present in the genomes.

## 2. Materials and Methods

### 2.1. Bacterial Strains

*S.* sp. HRI was isolated in 2018 from a DDAC-based disinfectant bottle (1% *v/v*) at room temperature at the University of the Free State in South Africa [9]. *S.* sp. HRI can be accessed on NCBI as BioProject PRJNA580358, BioSample SAMN13155787, and the DDBJ/ENA/GenBank database under accession number WIXF00000000. *S. marcescens* subsp. *marcescens* ATCC 13880 strain was obtained from the American Type Culture Collection in 2018 for use at the Veterinary Biotechnology Laboratory at the University of the Free State. *S. marcescens* subsp. *marcescens* ATCC 13880 can be accessed on NCBI as BioProject PRJNA716961, BioSample SAMN18473280, and the GenBank database under accession number CP072199.1. Bacterial isolates were obtained from the University of the Free State’s bacterial culture collection. The isolates were aerobically grown at 37 °C for 18–20 h in Tryptic Soy Broth (TSB) or on Tryptic Soy Agar (TSA) plates for routine bacterial cultivation.

### 2.2. Preparation of Disinfectants and Inhibitors

Benzalkonium chloride (BAC) (Sigma-Aldrich, St. Louis, MI, USA), didecyldimethylammonium chloride (DDAC), and HyperCide^®^ (14% peracetic acid (CH_3_CO_3_H) and 22% hydrogen peroxide (H_2_O_2_)) (ICA International Chemicals, Stellenbosch, ZAF) were used in this study. The disinfectants were freshly prepared in sterile distilled H_2_O before each experimental requirement. RSP (Sigma-Aldrich, St. Louis, MI, USA) and CCCP (Sigma-Aldrich, St. Louis, MI, USA) were used at working concentrations of 5 µg/mL and 12.5 µM, respectively.

### 2.3. Efflux Pump Predictions and Annotations

The whole genome sequences of *S.* sp. HRI and *S. marcescens* subsp. *marcescens* ATCC 13880 were obtained from the National Centre of Biotechnology Information (NCBI) and are accessible using the following identification codes: *S.* sp. HRI: [GenBank]: 29000988, [Taxonomy I.D]: 2663241 and *S. marcescens* subsp. *marcescens* ATCC 13880: [GenBank]: 25904888, [Taxonomy I.D]: 911022. The transporter profile of both isolates was assessed using the Transporter Automatic Annotation Pipeline (TransAAP) tool (http://www.membranetransport.org/transportDB2/index.html (accessed on 16 August 2022)), developed by Professor Ian Paulsen, Dr. Liam Elbourne, Dr. Karl Hassan and Dr. Sasha Tetu, 2017 [15]. The TransAAP tool utilizes a relational database to predict the cytoplasmic membrane protein complement of organisms whose whole genome sequences are available by inserting the taxonomic I.D. The transporter proteins of each organism were classified into protein families according to the transporter classification system, and substrate/function predictions were provided for each transporter protein. Each entry was individually curated to identify resistance functionality, such as antibiotic, disinfectant, multidrug or heavy-metal resistance. The identified resistance transporters were further verified using the NCBI BLAST https://blast.ncbi.nlm.nih.gov/Blast.cgi (accessed on 17 August 2022), UniProt BLAST (https://www.uniprot.org/blast/ (accessed on 17 August 2022)), and Kyoto Encyclopedia of Genes and Genomes (KEGG) BLAST (https://www.genome.jp/tools/blast/ (accessed on 17 August 2022)). The transporter identity was deemed accurate depending on the findings of the three BLAST results.

### 2.4. Susceptibility Testing

The disinfectant resistance profiles of *S. marcescens* subsp. *marcescens* ATCC 13880 and *S.* sp. HRI, were determined by minimum inhibitory concentrations (MICs) using the broth microdilution method so that the HRI isolate could be classified as resistant or susceptible compared to the ATCC strain. The disinfectants were chosen based on categorical requirements such as a first-generation QAC, a fourth-generation QAC, and a non-QAC. One to two colonies were picked and resuspended in PBS. These suspensions were adjusted to a 0.5 McFarland standard (around 10^8^ CFU/mL). A two-fold dilution range was prepared by serially diluting a stock concentration of disinfectant. Then, the bacterial suspensions were added to each diluted disinfectant solution in the range for the contact time. After the contact time of 20 min, each dilution was transferred to brain heart infusion (BHI) broth and incubated overnight. The MICs were determined as the lowest concentration that prevented visible growth. All MICs included at least three technical replicates and three biological replicates.

### 2.5. Efflux Pump Inhibitor-Supplemented Susceptibility Tests

The efflux pump activity of the *Serratia* isolates was assessed using RSP (Sigma-Aldrich, St. Louis, MI, USA) at a final concentration of 5 µg/mL and CCCP (Sigma-Aldrich, St. Louis, MI, USA) at a final concentration of 12.5 µM. These EPIs were added by using a broth microdilution method. One to two colonies of each strain were picked and resuspended in PBS. The suspensions were adjusted to a 0.5 McFarland standard. A two-fold dilution range was prepared by serially diluting a stock concentration of test disinfectant. Then, the EPIs were added to each dilution to make up the final concentrations, followed by bacterial suspensions to each serial disinfectant dilution. After the 20 min contact time expired, each dilution was transferred to BHI broth and incubated overnight. The EPI control consisted of the final concentration of each EPI and bacterial suspensions to ensure no intrinsic antibacterial activity at concentrations used in experiments. The MIC was determined as the lowest disinfectant concentration under the test that prevented visible bacterial growth in microcentrifuge tubes after incubation. All MIC experiments included at least three technical replicates and three biological replicates.

### 2.6. Statistical Analysis

The data presented in Figure 1 and Figure 2 depicts the frequency of various types of efflux pump superfamilies produced by a prediction pipeline. These data were analyzed using the chi-squared test on SAS 9.4. The data presented as bar graphs (Figure 3, Figure 4 and Figure 5) depicts the mean disinfectant concentrations ± standard deviation (*n* = 3 biological replicates) and was analyzed using SAS 9.4. Data from Figure 3 were compared using two-sample *t*-tests assuming equal/unequal variances based upon the *f*-test for the variance of two groups. *p* < 0.05 was considered statistically significant. Data from Figure 4 and Figure 5 were compared using a multiple comparison of samples using the Bonferroni-adjusted ANOVA. *p* < 0.05 was considered statistically significant.

## 3. Results

### 3.1. Efflux System Profile of Isolates

Efflux pump gene arrangements and efflux pump genes associated with antimicrobial resistance were identified in both *Serratia* isolates. Both isolates had representative transporters from all the efflux pump superfamilies, although the relative abundance of specific transporters varied (Figure 1). The efflux pump profile of *S. marcescens* subsp. *marcescens* ATCC 13880 displayed most of the identified transporters were from the ABC superfamily followed by the MFS family (Figure 1). The DMT RND and MOP superfamily transporters constituted a tiny portion of the total transporter repertoire (Figure 1). Finally, the greater portion of other transporters which were not classified into the five efflux superfamilies primarily consist of transporters for ion exchange amino acid transport and nutrient uptake/export. These include sugar phosphotransferase systems (PTS) the amino acid-polyamine-organocation (APC) family and the resistance to the homoserine/threonine (RhtB) family.

The efflux pump profile of *S*. sp. HRI was similar to the ATCC strain in that the vast majority of the displayed transporters belong to the ABC superfamily (Figure 1). Similarly, the second most abundant efflux transporter family was the MFS transporters (Figure 1). The DMT superfamily comprised a small proportion of the total transporter complement (Figure 1). Biocide-specific SMR subfamily transporters were present, such as QacE, SugE, and EmrE conferring elevated disinfectant tolerance. The MOP superfamily and the RND superfamily were represented the least in the total repertoire, lacking any biocide-specific transporters (Figure 1). A large proportion of transporters were unable to be classified into the five efflux superfamilies (Figure 1). Some of the more abundant transporters present in this grouping were from the APC family, the RhtB family, and the solute: sodium symporter (SSS) family. Many of the unclassified transporters function as transporters for molecules, such as threonine, serine, ions, and metals.

Resistance transporters in *S. marcescens* subsp. *marcescens* ATCC 13880 and *S.* sp. HRI constituted a small proportion of the total transporters identified (Appendix A). The ATCC strain had resistance transporters from all five efflux superfamilies (Figure 2). The MFS family represented the greatest proportion of resistance substrate/function efflux pumps. The DMT superfamily had the second-highest abundance of resistance efflux pumps, followed by the RND and the ABC superfamilies which shared an equal abundance of transporters implicated in resistance. The identified members of the RND superfamily were permease/membrane fusion subunits mostly predicted to have multidrug resistance functioning. Many of the ABC superfamily members were uncategorized according to the BLAST results. Furthermore, several transporters were uncharacterized, such as YbhF/YbhS, but these proteins had multidrug functioning predictions (Appendix A). The MOP superfamily consisted of a very small proportion of the transporters, mainly from the multidrug and toxin extrusion (MATE) subfamily. Identified transporters from the MATE subfamily included NorM and MdtK. The aromatic acid exporter (ArAE) family consisted of the largest proportion of transporters not grouped into the five efflux superfamilies. These transporters were mainly predicted to have fusaric acid transport functioning (Appendix A). The proteobacterial antimicrobial compound efflux (PACE) family is one of the two new transporter families that has recently been identified [16]. Two transporters from this family were predicted; however, no protein identity was elucidated from the amino acid sequences (Appendix A).

Interestingly, *S.* sp. HRI had fewer resistance transporters than the ATCC strain (Figure 2). However, the predicted resistance superfamily grouping ratio was similar for both isolates. The MFS family had the greatest abundance of resistance proteins in the HRI isolate, mainly consisting of transporters of the multiple drug transporter (Mdt) family (Appendix A). The DMT and RND superfamilies shared equal numbers of transporters and represent the second most abundant grouping of resistance proteins predicted. Mdt and RarD proteins were predicted to have multidrug and chloramphenicol substrate functioning, respectively (Appendix A). The ABC superfamily and the MOP superfamily composed a small number of the resistance transporters. The ABC superfamily was predicted to have multidrug function proteins, such as YbhR and YbhF. Additionally, a singular macrolide-specific protein, MacA, (Appendix A), and two MATE subfamily transporters, MdtK and EmmdR/YeeO, with multidrug substrate functioning were predicted (Appendix A).

### 3.2. Minimum Inhibitory Concentrations for Isolates

The mean disinfectant concentrations for MIC results are presented in bar charts to better visualize the differences between the *Serratia* isolates (Figure 3, Figure 4 and Figure 5). Susceptibility tests for BAC showed that the MIC for the HRI isolate was 49-fold higher than the ATCC strain. Thus, according to the criteria required to classify an organism as resistant, the HRI isolate is resistant to BAC. The addition of RSP increased the susceptibility of both *Serratia* isolates to disinfection (Figure 4 and Figure 5). The mean disinfectant concentration that the ATCC strain could tolerate was reduced 5-fold, and the HRI isolate disinfectant tolerance was reduced 42-fold with RSP supplementation (Figure 4 and Figure 5). Unexpectedly, the addition of CCCP caused significant increases in the mean disinfectant concentrations that the isolates could tolerate. BAC disinfection with CCCP supplementation resulted in a 2-fold and 4-fold increase in the mean disinfectant concentration tolerance for the ATCC and HRI isolates, respectively (Figure 4 and Figure 5).

The mean disinfectant concentration of DDAC the HRI isolate was able to tolerate was lower than the mean disinfectant concentration of BAC. However, the ATCC strain displayed higher tolerance toward DDAC than BAC. Like BAC disinfection, the supplementation of RSP increased the *Serratia* strains’ susceptibilities toward DDAC disinfection. The susceptibility towards DDAC increased 9-fold and 6-fold for the ATCC and HRI strains, respectively (Figure 4 and Figure 5). CCCP supplementation during DDAC disinfection increased susceptibility 4-fold against the ATCC strain (Figure 5). A paradoxical decrease in susceptibility was reported when DDAC + CCCP was used in combination testing against the HRI isolate. The mean disinfectant concentration that the HRI isolate tolerated increased 1-fold after CCCP supplementation (Figure 4).

The HRI isolate was able to tolerate a significantly higher concentration of HyperCide^®^ disinfectant than the ATCC strain, as depicted in Figure 3. RSP supplementation increased susceptibility toward HyperCide^®^, producing 4- and 11-fold decreases in the mean disinfectant concentration of the ATCC and HRI strains were able to tolerate, respectively (Figure 4 and Figure 5). The decline seen when RSP was added indicates that efflux pump activity is responsible, at least in part, for the tolerance toward HyperCide^®^ in both isolates. The addition of CCCP to HyperCide^®^ treatment caused mixed results for both isolates. CCCP supplementation increased the HRI isolate’s susceptibility and decreased the ATCC strain’s susceptibility to HyperCide^®^ disinfection (Figure 4 and Figure 5).

## 4. Discussion

Bacterial resistance against QACs and other disinfectants has been reported in many different bacterial species, including ESKAPE pathogens (*Enterococcus faecium*, *Staphylococcus aureus*, *Klebsiella pneumonia*, *Acinetobacter baumannii*, *P. aeruginosa* and *Enterobacter* sp.) [17,18]. However, resistance to disinfectants is not restricted to these species only, resistance has been observed in *Listeria monocytogenes* and *S. marcescens* in the food and medical industries, respectively [19,20]. Both chromosomal- and plasmid-encoded efflux pumps confer resistance against disinfectants [21,22]. Efflux machinery is well-adapted to reduce/prevent cellular damage in environments with hazardous chemicals and toxic metabolic waste products [23]. Active transport of any hazardous compounds out of the cell decreases the intracellular concentration of the compounds, allowing for increased cell survivability [21].

During this study, the transporters of each isolate were predicted using an annotation pipeline that uses the transporter classification system to provide an identity and substrate/function prediction for each transporter (Figure 1). Both isolates had few transporters with resistance substrate/function predictions (Figure 2). However, the numerical count of efflux pumps cannot be directly correlated to an expected resistance phenotype since the acquisition/activation of a single gene for an efflux pump or multidrug efflux pump can make a cell less susceptible [24].

The ATCC strain was found to have MFS efflux pumps, such as EmrB, MdtG, MdtL, and MdtD, implicated as multidrug transporters with antibiotic resistance capabilities [25]. The DMT superfamily resistance efflux pumps were mainly from the SMR subfamily, such as MdtJ, SsmE/QacE, and SugE (Appendix A). The SugE transporter was one of the three possible QAC-specific transporters to be identified [26]. The other potential QAC-specific transporters are QacE from the DMT superfamily and QacA from the MFS. These plasmid-borne efflux pumps are well-studied in *S. aureus* but can be spread to other species through horizontal gene transfer [27,28]. The RND superfamily had efflux pumps, such as AcrD, SdeB, MexW/MexI, and MdtB/MuxB, all with multidrug substrate/function (Appendix A). Most of the members of the ABC superfamily were uncharacterized in function according to BLAST, such as YbhF/YbhS (Appendix A). The MOP flippase superfamily consisted of a very small proportion of the transporters mainly from the MATE subfamily. This subfamily of transporters is well known to contribute to multidrug resistance with well-studied transporters such as MdtK, DinF, and NorM identified (Appendix A) [29,30]. Two transporters from the PACE family were predicted; however, the BLAST results could not provide the transmembrane protein identity. Generally, proteins from the PACE family transport biocides, such as chlorhexidine, suggesting these proteins could be involved in disinfectant tolerance seen in the ATCC 13880 strain [31].

Surprisingly, *S.* sp. HRI had fewer resistance transporters compared to the susceptible ATCC strain (Figure 2). The SMR subfamily proteins predicted include QacE, SugE, EmrE, and SsmeE. These proteins are well established to confer resistance to disinfectants, specifically QacE and SugE which have specificity for QACs (Appendix A) [26,32]. The *Serratia* isolates shared all the predicted QAC-specific efflux pumps, suggesting that these efflux pumps are not responsible for the higher level of resistance observed in *S*. sp. HRI compared to the type strain. A comparison of the efflux pumps revealed that the HRI isolate had two efflux pumps absent from the ATCC strain. These pumps are the SMR transporter EmrE and the MFS transporter SmfY. (Appendix A). EmrE has been shown to mediate QAC tolerance in *L. monocytogenes*, and SmfY is a multidrug efflux pump characterized in *S. marcescens* conferring tolerance to a range of antimicrobial agents [20,33]. Although these pumps could be responsible for the resistance of the HRI isolate, differential expression of the shared efflux pumps between the two isolates could also add to the vastly different levels of susceptibility.

SdeB and SdeY are members of the RND superfamily. Both efflux systems were predicted in both isolates. These efflux systems have been previously identified as part of the SdeAB and SdeXY systems in *S. marcescens* [24,34]. Ethidium accumulation assays using CCCP in *E. coli* clones carrying *sdeXY* genes showed that SdeXY is an energy-dependent drug efflux pump resulting from the inhibition that CCCP induced [24].

The MFS pump EmrB was predicted in both isolates. This monomeric efflux protein can form a tripartite system with EmrA and TolC; however, it was predicted to function exclusively in HRI [35]. The transcriptional repressor, EmrR, negatively regulates the expression of the *emrAB* operon by binding the promotor region [36]. There is limited knowledge of the functionality of EmrR to regulate *emrB* in its monomeric form. The substrate specificity of EmrAB-TolC has been shown for antibiotics, a set of unrelated antimicrobial substances that act as uncouplers of the PMF and hydrophobic compounds. CCCP is a substrate of EmrAB-TolC, which directly binds to EmrR to induce the expression of *emrAB* [37]. Therefore, if EmrB has substrate specificity for CCCP and EmrR regulates EmrB, the inactivation of EmrR would increase the functionality of EmrB reducing the uncoupling of the PMF. This activation and subsequent activity of EmrB could explain the decreased susceptibility/paradoxical growth seen when CCCP is supplemented.

The MICs were determined for QACs and HyperCide^®^ in the presence and absence of EPIs. The MIC results displayed statistically significant differences between the MIC of the HRI and ATCC strains (Figure 3). Classifying the HRI isolate as resistant based on testing against the closest related type strain. Standard EPIs were chosen based on functionality, with RSP blocking the transporter channels of certain efflux families, such as RND, MF, and ABC superfamilies [38]. The susceptibility changes show that some specific efflux pump activity is present in conferring disinfectant resistance/tolerance. RSP supplementation significantly reduced the MICs for all disinfectants tested (Figure 4 and Figure 5). From the results, efflux pump(s) from the ABC, MF, or RND superfamilies is/are responsible for QAC and HyperCide^®^ associated resistance.

The protonophore CCCP acts by uncoupling oxidative phosphorylation which disrupts the ionic gradient of bacterial membranes [39]. All the efflux superfamilies, excluding the ABC superfamily, utilize the PMF as part of the cellular metabolism [40]. The effect of CCCP has been shown to lower the MIC of various biocides tested, including BAC, and therefore increase susceptibility [41]. The same effect has been observed with ciprofloxacin as the addition of CCCP decreased MIC values [42]. However, this study showed significantly higher MIC values for *Serratia* species after the supplementation of CCCP to BAC and HyperCide^®^. Significantly these contradictory decreases in the susceptibility seen when CCCP is supplemented counters the theoretical stance that eliminating energy supply to the efflux pumps via the PMF will inactivate the proteins, thereby increasing the susceptibility to disinfection.

Similar results were observed in a study using a different efflux inhibitor phenylalanine-arginine β-naphthylamide (PAβN), which resulted in paradoxical effects on bacterial inhibition not related to solubility problems [43]. Similarly, in this study, no solubility problems were encountered during the supplementation of CCCP. The Eagle effect could be responsible if the loss of activity cannot be attributed to inconsistent solubility. The Eagle effect is defined as the paradoxical reduced killing of microorganisms at antibiotic concentrations higher than the optimal bactericidal concentration (OBC) unrelated to the solubility [44]. This effect has been described for Gram-positive, Gram-negative, and mycobacteria exposed to numerous antibiotic classes with diverse chemical structures, cellular targets, and sites of action [45,46,47]. Generally, cell wall-active agents are commonly associated with a paradoxical effect. Antibiotics such as penicillin inhibit protein synthesis to the extent that prevents growth necessary for the drug’s lethal effects to occur. Many reports describing the Eagle effect have implicated that the action of autolysins, β-lactamases, and reactive oxygen species (ROS) contribute to the outcome depending on the test antibiotic [48,49,50]. The Eagle effect has been described for antibiotic treatment and EPI-supplemented antibiotic treatment [51]. The Eagle effect has not yet been implicated in paradoxical reduced killing after EPI-supplemented disinfection.

In conclusion, the results demonstrate that the use of EPIs can be used to improve the in vitro susceptibility of *Serratia* strains to various disinfectants. This work shows that efflux pumps provide resistance to *S.* sp. HRI to test disinfectants within the given contact time and that specific superfamilies predicted can be linked to the resistance. Further investigation is required to determine if efflux pumps are involved in the long-term resistance capabilities of the *Serratia* strains. Additionally, further studies should evaluate the effect of these inhibitors in combination with other antimicrobials and their impact on different microorganisms. An interesting avenue for further study would be to couple the use of EPIs with favorable pharmacokinetics and toxicity profiles in addition to disinfectants in combination therapy techniques where MDR isolates are common. This combination therapy approach could have massive implications in battling MDR profiles in bacteria and the medical industry rife with “superbugs” resistant to many antimicrobials.

## Figures and Tables

**Figure 1 microorganisms-10-02257-f001:**
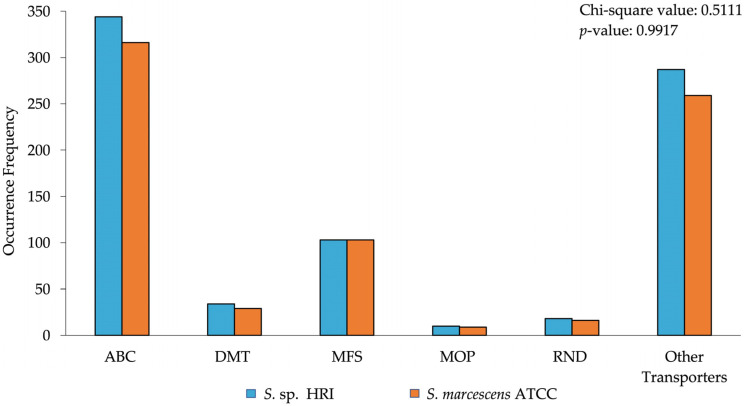
Total transporter predictions of *Serratia* isolates into five main superfamilies of efflux pumps. The *p*-value of the chi-square test > 0.05; therefore, the *Serratia* isolates, and occurrence frequency of efflux pump superfamilies are independent.

**Figure 2 microorganisms-10-02257-f002:**
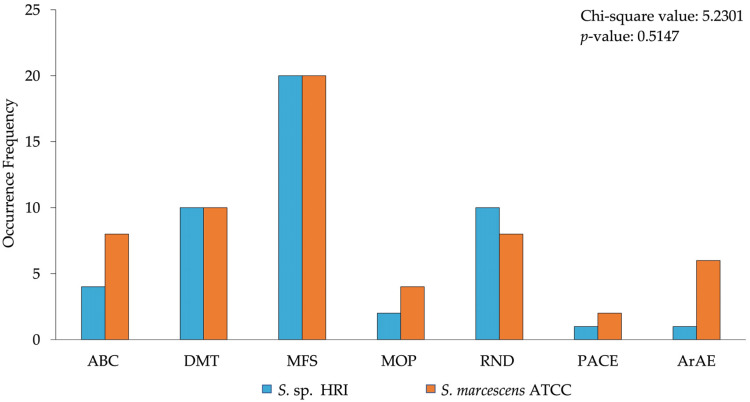
Resistance transporter predictions of *Serratia* isolates into various efflux families. The *p*-value of the chi-square test > 0.05; therefore, the *Serratia* isolates, and occurrence frequency of efflux pump families are independent.

**Figure 3 microorganisms-10-02257-f003:**
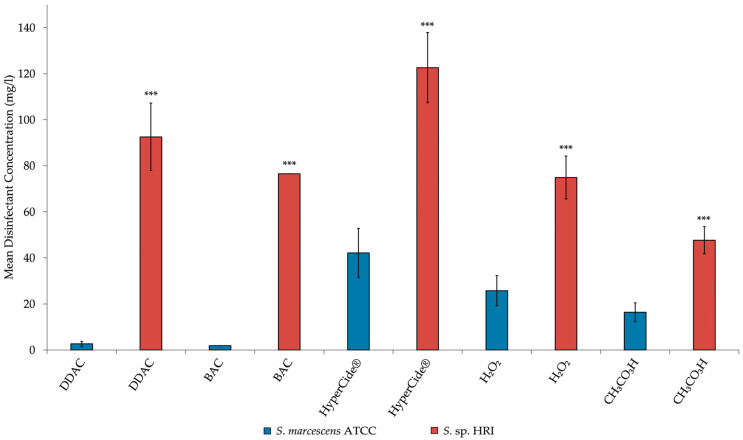
Mean disinfectant concentration of the MIC of *Serratia* isolates. Bars: geometric mean ± 95% C.I. as error bars (*n* = 3 biological replicates). Significance of difference to type strain indicated by asterisks: *** *p* < 0.001 (two-tailed unpaired *t*-test of disinfectant concentrations).

**Figure 4 microorganisms-10-02257-f004:**
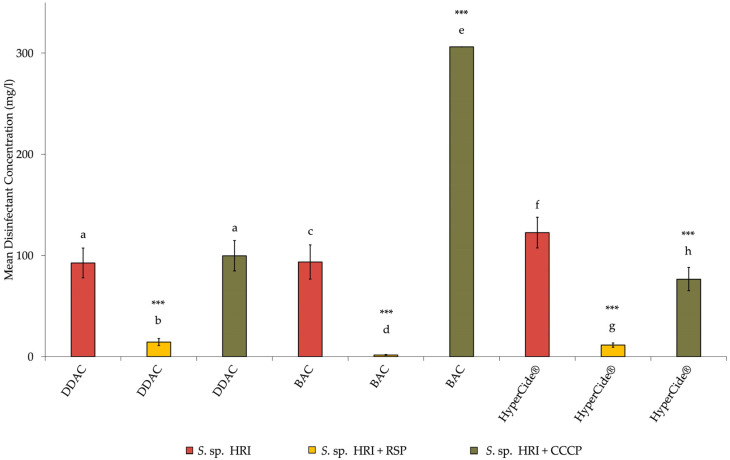
Mean disinfectant concentration of MIC for *S.* sp. HRI with supplementation of EPIs RSP at 5 µg/mL and CCCP at 12.5 µM final concentrations. Bars: geometric mean ± 95% C.I. as error bars (*n* = 3 biological replicates). Significance of difference to HRI isolate with only disinfectant indicated by asterisks: *** *p* < 0.001 (multiple comparison Bonferroni-adjusted ANOVA). Different letters indicate differences to HRI isolate with only disinfectant.

**Figure 5 microorganisms-10-02257-f005:**
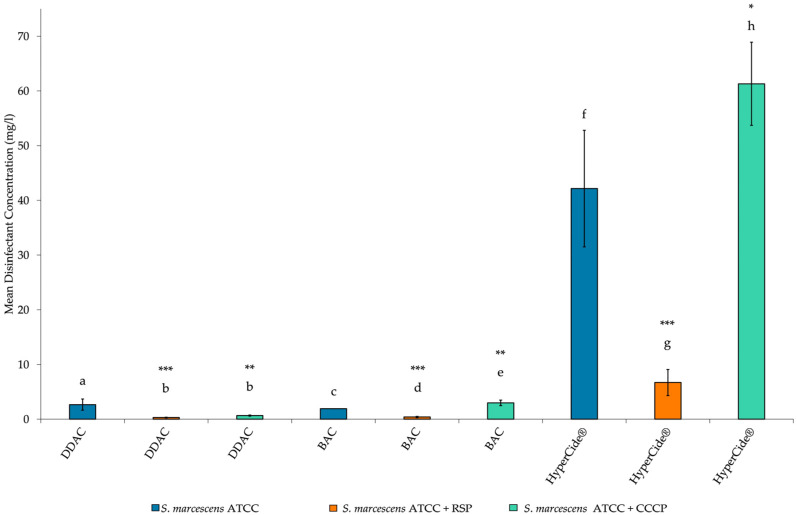
Mean disinfectant concentration of MIC for *S. marcescens* subsp. *marcescens* ATCC 13880 with supplementation of EPIs RSP at 5 µg/mL and CCCP at 12.5 µM final concentrations. Bars: geometric mean ± 95% C.I. as error bars (*n* = 3 biological replicates). Significance of difference to ATCC strain with only disinfectant indicated by asterisks: * *p* < 0.05; ** *p* < 0.01; *** *p* < 0.001 (multiple comparison Bonferroni-adjusted ANOVA). Different letters indicate differences to ATCC isolate with only disinfectant.

## Data Availability

Not applicable.

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
