# Peer review of "Susceptibility Tests and Predictions of Transporter Profile in Serratia Species"

_microorganisms, 2022, doi:10.3390/microorganisms10112257_

Round 1

Reviewer 1 Report

This paper is fully described with great work, well-designed and  well-presented and the authors demonstrated that the use of EPIs can be used to improve the in vitro susceptibility of Serratia strains to various disinfectants. This work also shows that efflux pumps function in providing resistance to Serratia sp. strain HRI to test disinfectants within the given contact time and that specific superfamilies predicted can be linked to the resistance.

It is my pleasure to recommend it for publication.

In the final draft, the authors should enlarge y axis title for figures 3,4,5 and add x titles (Enlarge all data inside the figures to be clear to the respective reader)

Author Response

Point 1: In the final draft, the authors should enlarge y axis title for figures 3,4,5 and add x titles (Enlarge all data inside the figures to be clear to the respective reader).

Response 1: The text in all the figures was enlarged to match the main text size in the article. Additionally, a legend describing each of the data entries was added beneath each figure.

The article was submitted for language and style revisions.

Thank you for the invaluable critique of the reviewer, all points were addressed to the best of all the authors’ knowledge and abilities.

Reviewer 2 Report

Abstracts.

First appearance of "HRI." Spell it out, please.

The text in Figures 3, 4, and 5 is too small to read!

For Figure 4 and 5, please put the name of the series directly below each bar chart.

Concentrations of reserpine and other inhibitors used

If there is a concentration-dependent inhibitory effect, we should consider using different concentrations. Also, is it correct to assume that all of these inhibitors are removed after bacteria have been sterilized? Is it assumed that they will be used for drinking water after disinfection?

Have you confirmed inhibition (using positive or negative controls) using typical transporter inhibitors?

If "Disinfenctant concentration" on the vertical axis in the figure indicates MIC, then it should be MIC.

Author Response

Point 1: Abstracts

Response 1: Changes made to abstract to clarify outcomes and results.

Point 2: First appearance of "HRI." Spell it out, please.

Response 2: First appearance of “HRI” changed to suit reviewers’ recommendations.

Point 3: The text in Figures 3, 4, and 5 is too small to read!

Response 3: Text size in all figures increased for better visualisation.

Point 4: For Figure 4 and 5, please put the name of the series directly below each bar chart.

Response 4: Series inserted directly below each Figure in the article.

Point 5: Concentrations of reserpine and other inhibitors used.

Response 5: Final concentrations of reserpine and CCCP added to methods section under “Efflux pump inhibitor supplemented susceptibility tests” according use in experimental methods. Additionally, the final concentrations of the inhibitors used added to the figure descriptions of Figure 4 and 5.

Point 6: If there is a concentration-dependent inhibitory effect, we should consider using different concentrations. Also, is it correct to assume that all of these inhibitors are removed after bacteria have been sterilized? Is it assumed that they will be used for drinking water after disinfection?

Response 6: The concentrations of inhibitors used in the “Efflux pump inhibitor supplemented susceptibility tests” did not exert any inhibitory effects, assessed by testing various concentrations of inhibitors to determine at which concentrations inhibition occurs. Additionally, the inhibitor control (bacteria + desired inhibitor at final concentration described) presented the same growth as the positive control. This was a vital component of the experimental design to ensure that no inhibition seen in bacterial growth is as a result of inhibitors themselves. The 2 ml Eppendorf tubes in which the MIC tests were conducted were sealed and discarded into biohazard containers, which were processed by a contracted biological waste removal company to ensure proper handling of the hazardous materials.

Point 7: Have you confirmed inhibition (using positive and negative controls) using typical transporter inhibitors?

Response 7: Only two efflux pump inhibitors were used to assess inhibitory effects. These two inhibitors were chosen based on literature investigation for standard inhibitors in efflux pump activity testing. Additionally, the functionality of these inhibitors differs, therefore they are able to target different transporters. In this way, all the efflux pump superfamily were inhibited by the supplementation of reserpine and carbonyl cyanide 3-chlorophenylhydrazone.

Point 8: If “Disinfectant concentration” on the vertical axis in the figure indicates MIC, then it should be MIC.

Response 8: Figure 3-5 y-axis descriptions altered to “mean disinfectant concentration”, to suggest that the mean disinfectant concentration of each bar graph is the MIC according to this experiment.

The article was submitted for language and style revisions.

Thank you for the invaluable critique of the reviewer, all points were addressed to the best of all the authors’ knowledge and abilities.

Reviewer 3 Report

The topic of the article is relevant, but the results cannot be considered reliable due to the lack of correct statistical processing.

1. The content of the "Statistical analysis" subsection indicates the incorrectness of statistical processing. The authors did not adjust for multiple sample comparisons. The authors did not use multiple sample comparison methods.

2. Figures 1 and 2 should not be presented in such a form. Obviously, the authors expect the reader to compare the right and left parts of the figure. The similarity of these parts does not guarantee the absence of significant differences between them. I recommend the authors to present the data in the form of a table or histogram columns. In this case, each part of the figure which is highlighted in one color should be located in pairs side by side. It is also necessary to apply the correct method of comparing the occurrence frequency (for example, the chi-square test). Reliability of differences between sectors that are filled with one color should be indicated by one, two or three asterisks on the corresponding columns of the histogram.

3. The reliability of the data in Figure 3 raises doubts: why are the +- standard deviations the same in all columns of the histogram? According to the theory of probability, this is impossible. The title of this figure does not indicate that the columns represent mean values, and + is the standard deviation. The repetition of experiments is not indicated.

4. In figures 3, 4, 5 there is no correct indication of the results of comparing the samples with each other. There, it is would be good using some method of multiple comparison of samples, such as the Tukey's HSD test.

5. In Figure 5, the +- standard deviation in all columns is proportional to their height. This is questionable as well.

5. The size of the text and numbers in all figures should be commensurate with the font size of the main text of the article.

6. It would be better not to duplicate the numbers indicated in the figures in the text. At the very least, this duplication should be avoided as much as possible.

7. The "Results" section should not contain elements of description of methods or interpretation of results.

8. If the Serratia genus (or another genus of bacteria) is repeatedly mentioned in the text or title of figure (for example, Fig. 5), it should be shortened.

Author Response

Point 1: The content of the "Statistical analysis" subsection indicates the incorrectness of statistical processing. The authors did not adjust for multiple sample comparisons. The authors did not use multiple sample comparison methods.

Response 1: The statistical processing was adapted as recommended to suit required statistical requirements to improve reliability of results. The statistical processing was adapted to incorporate multiple sample comparison methods, such as Tukey’s test for data depicted in Fig. 4 and 5. Additionally, the chi-square test was used to analyse the data depicted in Fig. 1 and 2. The chi-square test showed that the categories in both Figure were found to be independent of one another.

Point 2: Figures 1 and 2 should not be presented in such a form. Obviously, the authors expect the reader to compare the right and left parts of the figure. The similarity of these parts does not guarantee the absence of significant differences between them. I recommend the authors to present the data in the form of a table or histogram columns. In this case, each part of the figure which is highlighted in one color should be located in pairs side by side. It is also necessary to apply the correct method of comparing the occurrence frequency (for example, the chi-square test). Reliability of differences between sectors that are filled with one color should be indicated by one, two or three asterisks on the corresponding columns of the histogram.

Response 2: Figures 1 and 2 were edited to present the data according to the reviewer’s recommendations, allowing proper highlighting of differences. Additionally, the correct statistical testing was used to analyse the data. Both categories in Figures 1 and 2 were independent from one another.

Point 3: The reliability of the data in Figure 3 raises doubts: why are the +- standard deviations the same in all columns of the histogram? According to the theory of probability, this is impossible. The title of this figure does not indicate that the columns represent mean values, and + is the standard deviation. The repetition of experiments is not indicated.

Response 3: The standard deviations of Figure 3 was revised. The title of the figure was adapted to indicate that the mean disinfectant concentration for each strain for each disinfectant. The experimental repetitions and statistical details were added to the figure description.

Point 4: In figures 3, 4, 5 there is no correct indication of the results of comparing the samples with each other. There, it is would be good using some method of multiple comparison of samples, such as the Tukey's HSD test.

Response 4:The comparison of samples was assessed using the recommended statistical test to correctly compare results. The main requirement was to show that statistically significant differences are present between sample without efflux pump inhibitors and samples with efflux pump inhibitors. Tukey’s test was used to assess the data in Fig 4 and 5.

Point 5: In Figure 5, the +- standard deviation in all columns is proportional to their height. This is questionable as well.

Response 5:The standard deviation of Figure 5 was revised ensuring the reliability. The two-fold dilution property of the MIC determination suggests that at higher disinfectant concentrations differences between dilutions becomes numerically greater. Additionally, Figure 3 and 4 standard deviations were also revised.

Point 6: The size of the text and numbers in all figures should be commensurate with the font size of the main text of the article.

Response 6: The size of the figure text and numbers were altered to match the main text size of the article.

Point 7: It would be better not to duplicate the numbers indicated in the figures in the text. At the very least, this duplication should be avoided as much as possible.

Response 7: Any duplicate numbers in the figures present in the text was removed.

Point 8: The "Results" section should not contain elements of description of methods or interpretation of results.

Response 8: Elements of methods and interpretation present in the “Results” section was removed.

Point 8: If the Serratia genus (or another genus of bacteria) is repeatedly mentioned in the text or title of figure (for example, Fig. 5), it should be shortened.

Response 8: All repeated instances of the Serratia genus beside the first were shortened with exception to when the Serratia strains were being referred to collectively.

The article was submitted for language and style revisions.

Thank you for the invaluable critique of the reviewer, all points were addressed to the best of the all the authors’ knowledge and abilities.

Round 2

Reviewer 2 Report

I have no additional comments.

Reviewer 3 Report

The article has improved a lot. However, there were shortcomings.

1. Figures 4 and 5 should have Bonferroni-adjusted ANOVA in their titles, not Tukey's test, whose results are displayed in different letters above the bars.

2. After mentioning the name of the computer program in brackets, you must specify the name of the company, country and year (line 169).

3. It is not clear why the authors chose a difference level of 0.0001. Standard are 0.05, 0.01 and 0.001. At least (line 167-175), this should be justified in detail in the text.

4. In a pairwise comparison (for example, in Figure 3), the control is usually indicated first, and the experiment is second. Then the designation of the reliability of differences is indicated above the experiment. You must specify a specific difference confidence value (for example, p = 3.2*10-14 superscript), and not be limited to *<0.0001 (in any case, * must be removed).

5. It is not necessary to capitalize all words in the titles of articles.

After eliminating these shortcomings, the article can be published.

Author Response

Point 1: Figures 4 and 5 should have Bonferroni-adjusted ANOVA in their titles, not Tukey’s test, whose results are displayed in different letters above the bars.

Response 1: The Tukey’s test for the data in Figure 4 and 5 was assessed using the Bonferroni-adjusted ANOVA according to the reviewer’s recommendations. Different results displayed in Figure 4 and 5 were indicated using different letters.

Point 2: After mentioning the name of the computer program in brackets, you must specify the name of the company, country and year (line 169).

Response 2: After mentioning the TransAAP tool program the developers of the program were named, the year of development and the reference to the article was added after mentioning it in the text.

Point 3: It is not clear why the authors chose a difference level of 0.0001. Standards are 0.05, 0.01, 0.001. At least (line 167-175), this should be justified in detail in the text.

Response 3: The difference levels chosen by the authors were changed to the standard levels to comply with reviewer’s recommendations.

Point 4: In a pairwise comparison (for example, in Figure 3), the control is usually indicated first, and the experiment is second. Then the designation of the reliability of differences is indicated above the experiment. You must specify a specific difference confidence value (for example, p = 3.2*10-14 superscript), and not be limited to *<0.0001 (in any case, * must be removed).

Response 4: Figure 3 was adjusted to indicate the control first, followed by the experiment samples. Additionally, the reliability of differences confidence values were removed as this information adds no additional information to accept or reject the null hypothesis.

Point 5: It is not necessary to capitalize all words in the titles of articles.

Response 5: The capitalization of all words besides the first and bacterial genus in title was removed. Additionally, any unnecessary capitalization of headings in article were also remove.

The article was submitted for language and style revisions using Peerwith platform and Grammarly software.

Thank you for the invaluable critique of the reviewer, all points were addressed to the best of the all the authors’ knowledge and abilities to ensure the best quality for publishing.
